# Improving Sequential Model Editing with Fact Retrieval

**Xiaoqi Han**♣    **Ru Li** ♣*    **Hongye Tan** ♣
**Yuanlong Wang** ♣    **Qinghua Chai** ♣    **Jeff Z. Pan**♠*
♣ School of Computer and Information Technology, Shanxi University, China
♠ ILCC, School of Informatics, University of Edinburgh, UK
♣ 202112407003@email.sxu.edu.cn,{liru,tanhongye,ylwang,charles}@sxu.edu.cn
♠http://knowledge-representation.org/j.z.pan/

## Abstract

The task of sequential model editing is to fix erroneous knowledge in Pre-trained Language Models (PLMs) efficiently, precisely and continuously. Although existing methods can deal with a small number of modifications, these methods experience a performance decline or require additional annotated data, when the number of edits increases.

In this paper, we propose a **R**etrieval **A**ugmented **S**equential Model **E**diting framework (**RASE**) that leverages factual information to enhance editing generalization and to guide the identification of edits by retrieving related facts from the fact-patch memory we constructed. Our main findings are: (i) State-of-the-art models can hardly correct massive mistakes stably and efficiently; (ii) Even if we scale up to thousands of edits, RASE can significantly enhance editing generalization and maintain consistent performance and efficiency; (iii) RASE can edit large-scale PLMs and increase the performance of different editors. Moreover, it can integrate with ChatGPT and further improve performance. Our code and data are available at: https://github.com/sev777/RASE.

## 1 Introduction

Pre-trained Language models (PLMs) are trained on massive amounts of texts, encoding knowledge in their parameters, and having remarkably succeeded in knowledge-driven tasks such as question answering (Kwiatkowski et al., 2019; Chen et al., 2021, 2022; Hu et al., 2023) and reasoning (Mihaylov et al., 2018; He et al., 2023). Such parametric knowledge complements (Pan et al., 2023) the explicit and structured knowledge in widely used knowledge graphs (Pan et al., 2017a,b).

As Pre-trained Language models are deployed widely, the need to keep their parametric knowledge correct and up-to-date without massive retraining costs becomes increasingly important (Sinitsin

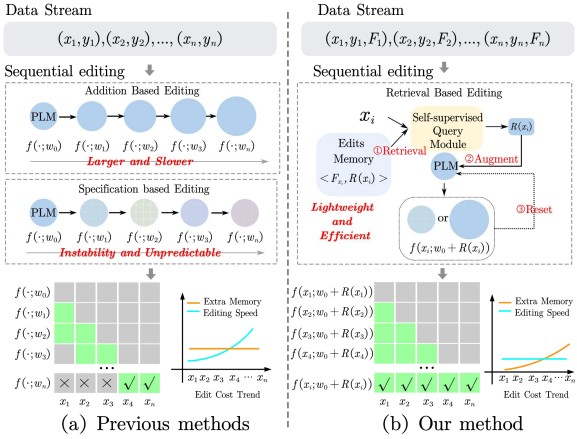

Figure 1: Comparison of RASE with other methods. Other methods (a) involve continuous modification of PLM's parameters. However, the efficiency of modification decreases as the number of edits grows, resulting in poor performance for SME. Our method (b) leverages an fact retrieval framework, guaranteeing consistent modification efficiency regardless of the number of edits. This improvement enhances both the scalability and performance of SME. Furthermore, by leveraging factual information relevant to editing, our retrieval method better adapts to SME scenarios.

et al., 2019). Prior works have proposed Model Editing (De Cao et al., 2021), enabling fast, data-efficient PLMs updates without damaging model performance on unmodified data. These methods focus on simultaneously modifying edits; however, the errors in PLMs are unpredictable and pervasive, thus correcting them at once is impractical.

To this end, Sequential Model Editing (SME) (Huang et al., 2023) has been proposed to fix a series of bugs in order while maintaining the previous edits and performance across unmodified data. This is a challenging task, since, for a highly non-linear model, even slight perturbations might significantly alter the model's output. As shown in Figure.1(a), prior works on SME make modifications by either directly modifying the parameters of the lan-

---
*Contact Authors

guage model (Meng et al., 2022, 2023) or continuously adding parameters to the model (Huang et al., 2023), and they use additional storage to maintain the model's performance on unmodified data (Locality). They have shown promise only to deal with a small number of modifications, and suffer from insufficient expressiveness: 1) As the number of edits increases, the model's parameters undergo significant changes, or the number of parameters becomes large, resulting in forgetting previous edits or unmodified data and gradually increasing the cost of modifications. 2) In SME, the edits may clash with pre-sampled data stored to preserve Locality. Thus, the cost of maintaining memory will increase when dealing with more edits.

One feasible way to address the abovementioned issue is the memory-based method, which uses an edit-history memory while keeping the original model frozen. However, the existing method (Mitchell et al., 2022b) primarily focuses on batch editing, assuming that all editing data is known and utilizing the data to train models for data-type identification and modification, which makes it inefficient when handling Sequential Model Editing.

To fully leverage the advantages of the memory-based method, we propose **RASE**, a **R**etrieval **A**ugmented **S**equential Model **E**diting framework, (cf.Figure.1(b)) stores edits with their fact description and parameter shift on PLMs in memory and uses a query module to retrieve from them to apply each modification individually. With this framework, we can simplify complex continuous modifications by breaking them down into multiple individual edits, and the SME can be decomposed into the following two sub-tasks: 1) *Edits classification*: determining whether the input needs modification. We propose a fact-aware contrastive model to identify edits without pre-training on any annotated datasets by learning the sentence and fact embedding in a self-supervised way. 2) *Editing*: determining how to efficiently modify each edit. We enhance the generalization capabilities of existing editors by incorporating factual information with the editing data.

Experimental results on fact-checking and question answering tasks indicate that RASE can rectify a series of mistakes (up to thousands) while retaining the model's performance on unmodified data. Our contributions include:

- We propose **RASE**, a retrieval augmented knowledge editing framework, constructing a

fact-patch memory, and using a query module to classify the edits by retrieving related facts from the memory and achieving efficient, stable, and expansible SME.

- Our method can be integrated with other factual knowledge editors, enhancing the generalization capability of each editor by leveraging facts related to the edits, and achieving more reliable modifications.

- Experiments show that RASE can support large PLMs for stable and efficient continuous editing. Moreover, we utilize ChatGPT to re-rank retrieval results, further enhancing the accuracy of fact retrieval and identify modified and unmodified data more accurately, thus maintaining the model's performance on the unedited data better.

## 2 Background

### 2.1 Model Editing

The task of Model Editing (ME) (De Cao et al., 2021) is to intervene the target model's behaviour on a specific example while preventing the model from forgetting unmodified data. Previous research can be classified into two categories:

**Specification-based Methods** These methods (Zhu et al., 2020; De Cao et al., 2021; Mitchell et al., 2022a; Meng et al., 2022, 2023; Han et al., 2023) fix the bugs through locating and modifying specific parameters in PLMs. However, a minimal change in parameter space may produce a completely different output for many data, which may leave post-edit model potentially unreliable.

**Addition-Based Methods** Instead of directly modifying the parameters of PLMs, the Addition-Based Methods (Dong et al., 2022; Mitchell et al., 2022b; Huang et al., 2023) utilize an additional module to apply the edits while keeping the parameters of the PLMs fixed. They allow better preservation of the original language model's performance and exhibits excellent scalability.

However, both types of methods focus on batch editing. In practice, language model errors often require timely and sustainable correction, and the to-be-modified data is unknown.

### 2.2 Sequential Model Editing

The task of Sequential Model Editing (SME) (Huang et al., 2023) is to fix a series of mistakes of a target model as soon as they appear.

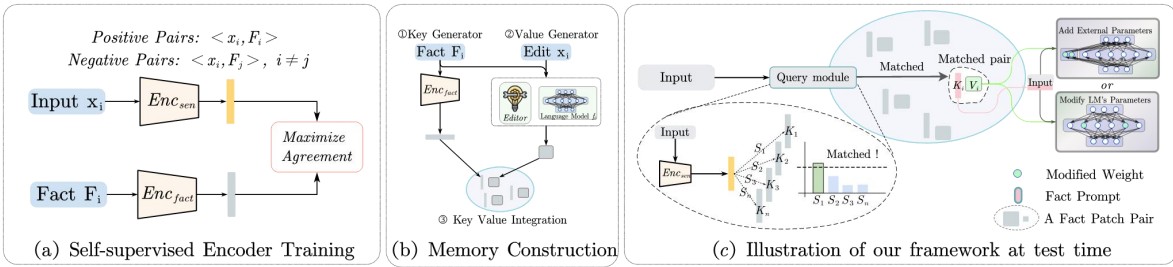

Figure 2: Illustration of RASE. Figure (a) illustrates the sentence encoder and fact encoder trained in a self-supervised way. Figure (b) shows the memory construction process. We use the facts embeddings as the key and the parameter shift for edits as the value. Figure (c) presents the retrieval framework, where the query module retrieves information from the memory for each input. Based on the retrieval results, we make appropriate modifications to the PLM, correcting the data that needs to be modified.

Formally, given a PLM $\mathcal{F}(\cdot)$ and an edit stream $\mathcal{D}\{(x_1, y_{x_1}), (x_2, y_{x_2}), \ldots, (x_n, y_{x_n})\}$ with $n$ edit inputs, the task is to effectively correct the output of $\mathcal{F}(x_i)$ when $\mathcal{F}(x_i) \neq y_{x_i}$, while maintaining accurate predictions for previous edits and unmodified data. We denote $\mathcal{E}$ as the editing function, after processing the $i\text{-}th$ input, the $\mathcal{F}_i(\cdot)$ can be represented as:

$$\mathcal{F}_i(\cdot) = \begin{cases} \mathcal{F}_{i-1}(\cdot) & \mathcal{F}_{i-1}(x_i) = y_{x_i}, \\ \mathcal{E}(\mathcal{F}_{i-1}(\cdot), x_i, y_{x_i}) & \mathcal{F}_{i-1}(x_i) \neq y_{x_i}. \end{cases} \quad (1)$$

Compared with ME, the desiderata for the SME method are as follow: (1) **Reliability**, the editor is supposed to successfully edit data sequentially, and the post-edit model should retain the output for previous edits after editing every edit; (2) **Generality**, the editor should generalize over the equivalent inputs[1] about the edits; (3) **Locality**, the editor should retain its accuracy on unmodified data.

Existing editing methods (Meng et al., 2022; Mitchell et al., 2022b; Huang et al., 2023) in Sequential Model Editing (SME) still exhibit limitations regarding the generalization and the scale of edits, moreover, due to their direct modify the parameters of the PLMs, the performance of locality may also become unreliable. To address it, we propose a retrieval-augmented editing framework, leveraging factual information associated with the edits as guidance and enhancing the generalization capability of the editor, leading to stable and scale sequential editing, furthermore, since we will keep the original PLMs frozen, and only make corre-

sponding changes to the PLMs based on the retrieval results, the performance of the retriever can ensure the reliability of the model's locality.

## 3 Approach

Figure 2 shows the overview of RASE. In a nutshell, we first train a fact encoder and a sentence encoder in a self-supervised way to maximize the similarity between sentences and their corresponding fact description. After applying each edit successful, we encode the corresponding fact as a key (Section 3.1.1), treat the required parameter shift as a value (Section 3.1.2), and store the (key,value) pair. During the evaluation, we compare a given input with the keys in the memory (Section 3.2). If a key is found, it indicates that the data needs modification, and the value will be applied to the language model to complete the editing. Otherwise, the input is simply passed to the original model. We illustrate our approach by introducing the facts-patch memory's construction and usage.

### 3.1 The facts-patch memory

The memory $M = (\mathbf{K}, \mathbf{V})$ contains facts representation $\mathbf{K}$ and the edit operation $\mathbf{V}$, with each key $k_i$ mapping to one value $v_i$. When an input has the highest similarity to any key in memory, we can consider that the input needs modification. Therefore, we only need to learn the embeddings that can maximize the similarity between the input and its relevant facts in order to achieve data classification for editing without relying on labeled data.

### 3.1.1 Construction of Key K

We use the key $\mathbf{K} = \{k_0, k_1, ..., k_i, ...\}$ to determine whether the input needs editing during evaluation. Each $k_i$ is generated by the fact encoder

---

[1]The inputs with the same meaning can differ in natural language expressions called equivalent inputs; e.g., 'Michael Jordan was born in ?' vs. 'The birthplace of Michael Jordan is ?'.

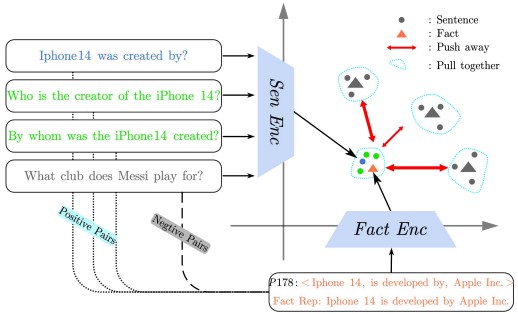

Figure 3: Fact-aware contrastive learning method: we represent the inputs (in blue), their equivalents (in green), and corresponding facts (in orange) as positives, while representing other facts and sentences (in gray) as negatives. Our goal is to pull together the positive pairs and push away the negative pairs by training two Encoders.

$Enc_f(\cdot)$, which aims to maximize the similarity between $k_i$ and the relevant sentences. Among many possible implementations of the encoder, a straightforward way is to utilize a language model or an existing retriever Izacard et al. (2022a) to encode the data. However, the representation is ineffective for cases where sentences and factual descriptions are distinct but semantically similar, especially for some sentences where only one word is different.

Thus, we propose a fact-aware contrastive learning method to training the encoder. Specifically, we training two Encoders $Enc_f(\cdot)$ and $Enc_s(\cdot)$ to get the fact embeddings $E_f$ and sentence embeddings $E_s$ respectively, and the goal is to maximize the similarity between the embeddings of sentences and factual information. As shown in Figure 3, for the fact "<Iphone 14, is developed by, Apple Inc.>", we view "Iphone 14 is developed by Apple Inc." as a factual description and use $Enc_f(\cdot)$ to encode it. Then, we use $Enc_s(\cdot)$ to encode its corresponding sentences as positive pairs (indicated in green and blue.) and unrelated sentences (indicated in grey.) as negative pairs. We trained $Enc_f(\cdot)$ and $Enc_s(\cdot)$ using the following three contrastive losses:

**Facts Contrastive Loss** ($L_{F2F}$). We use $L_{F2F}$ to maximize the similarity between the fact embedding $E_f$ and its positive embedding $E_f^{dp}$ ($dp$ means it is generated by dropout noise within transformer layers), and minimize the similarity between the fact and other facts. In this way, fact embedding is learned in a self-supervised way.

**Sentences Contrastive Loss** ($L_{S2S}$). $L_{S2S}$ and $L_{F2F}$ are computed in the same way. Since one fact corresponds to multiple positive sentences, so the average representation of these sentences is

viewed as the sentence embedding $E_s$ of $L_{S2S}$.

**Contrastive Loss between Sentence and Fact** ($L_{S2F}$) pulls the embeddings of sentences and their related fact close while keeping unrelated facts apart. To ensure that positive samples' similarity surpasses a predefined threshold, we use a margin-based similarity to better discriminate distinct but semantically similar pairs by increase the similarity between negative samples and decrease the similarity between positive samples. In summary, our total loss is:

$$L_{cl} = \lambda_1 L_{F2F} + \lambda_2 L_{S2S} + \lambda_3 L_{S2F}, \quad (2)$$

where $\lambda_i$ is hyperparameter. See more details in Appendix A. Notably, enable to enhance the generalization of editor, the factual description is stored simultaneously with the fact embedding in $M$ without labels. e.g. for the sentence "IPhone 14 was created by?", the factual description is: "IPhone 14 ‖ iPhone 14 is developed by ‖ IPhone 14 was created by?" During the evaluation, each input will be encoded by $Enc_s(\cdot)$ and computed with **K** to determine whether a modification is required.

### 3.1.2 Construction of Value $v_i$

The value $v_i$ for each key $k_i$ represents the edit operation for the edit $x_e$. In this paper, we choose an addition-based editor, T-Patch(Huang et al., 2023) and a specification-based editor ROME (Meng et al., 2022), as the base editor due to their efficiency and the ability to continue editing.

**T-Patch** (Huang et al., 2023) only requires training a certain number of neurons for each edit and inserts them at a designated layer of the Transformer. In contrast to Huang et al. (2023), we does not require additional memory to store training data to satisfy locality. Moreover, we concatenate the factual description with the input as a prompt to improve the generalization of the editor. For T-Patch, the value $v_i$ is the extra neurons.

**ROME** (Meng et al., 2022) locates the knowledge required modification with a key-value pair in one of the middle MLP layers and modifies the corresponding knowledge by directly updating the key-value pair. For ROME, we use factual description as a prompt to improve the generalization of the editing process. The value $v_i$ consists of vector value $v_*$, lookup key $k_*$ and a shared matrix $C$. For both two editors, we also store the edits simultaneously with the $v_i$ in $M$ to help to identify the data type. The details of these two editors are in

Appendix B. We maintain the $M$ during training in real-time. See more details in Appendix C.

## 3.2 The usage of memory

In this section, we introduce how to utilize memory $M$ to identify which data requires modification and achieve sequential model editing.

For each data, after we obtain the embedding of the input $x$ using the sentence encoder $H = Enc_s(x)$, we match the $H$ with the keys $\mathbf{K}$ in the memory $M$ by the score function:

$$\text{score} = \text{Cos}(H, \mathbf{K}), \tag{3}$$

where $Cos(\cdot)$ is the cosine similarity, $\mathbf{K}$ means the fact embedding matrix in memory $M$.

We then select the Top-K scores from $score$ and calculate the following conditions: (1) MAX: the highest similarity score; (2) DIFF: the difference between the top-1 and top-2 score; (3) STD: the standard deviation of the Top-K scores. When the following conditions are met, the data is the edit:

$$\text{ED} = \mathbb{1}_{(\text{MAX}>t) \text{ OR } ((\text{DIFF}>t_d) \text{ AND } (\text{STD}>t_s))}, \tag{4}$$

where $t, t_d$ and $t_s$ are the thresholds of the similar score, difference score and standard deviation value respectively.

In Eq. 4, for data with maximum score less than $t$, we make judgments based on the distribution of the score, such as STD and DIFF. However, it is difficult to determine whether they need to be modified based on their distribution when they are similar but different. Therefore, we propose the two-way score to enhance the differentiation between data points and improve the identification of edits.

**Two-Way Score** incorporates the similarity between the edits in the Top-K data and the input data. The final score is:

$$\text{score}_t = \text{score} + \text{Cos}(H, H_i), i \in k, \tag{5}$$

where $H_i$ means the edit embedding calculated by $Enc_s(x)$ for the edits corresponding to the Top-K fact. Finally, we re-select the Top-K scores from $score_t$ and use Eq.4 to identify the edits.

## 4 Experiments

### 4.1 Dataset and evaluation metrics

We evaluate RASE on fact-checking dataset FEVER (Thorne et al., 2018a) and Zero-Shot Relation Extraction (ZsRE) dataset (Levy et al., 2017).

We use the same data splits for both datasets as Huang et al. (2023), specifically we employ the original validate as $\mathbb{D}_{test}$ and the original $\mathbb{D}_{Train}$ is split into an edit set $\mathbb{D}_{edit}$, a new training set $\mathbb{D}'_{train}$ and a new validation $\mathbb{D}_{val}$. We denote the initial model as $f_0$ which is trained on $\mathbb{D}'_{train}$ and validated on $\mathbb{D}_{val}$. The model $f_0$ is sequentially edited on $\mathbb{D}_{edit}$. Supposing that there are $T$ total mistakes in $\mathbb{D}_{edit}$, $I(\cdot)$ represents the indicator function. After editing the $t$-th edit example $(x_t, y_{x_t})$, we obtain a post-edit model $f_t$, and we use the following metrics to evaluate our method:

**Success Rate** (SR): For each edit, we test if the post-edit model $f_t$ produces the desired prediction.

**Edit Retain Rate** (ER): After editing $T$ edits, we evaluate how many past edits are retained by the final model $f_T$.

**Generalization Rate** (GR): After editing $T$ edits, we evaluate if the post-edit model $f_T$ is edited successfully on the equivalents of the edits in $\mathbb{D}_{edit}$.

**Training Retain Rate** (TrainR): After editing $T$ edits, we compare the performances of the post-edit model $f_T$ and the initial model $f_0$ on sub-dataset $\mathbb{D}_{tr}$ which is randomly sampled from $\mathbb{D}'_{train}$.

**Test Retain Rate** (TestR): After editing $T$ edits, we compare the performances of the post-edit model $f_T$ and the initial model $f_0$ on original validate dataset $\mathbb{D}_{test}$.

### 4.2 Baseline and Experiment Details

We compare RASE with the following knowledge editors: (1) the specification-based methods: Fine-Tuning (FT) (Zhu et al., 2020), FT with KL divergence (FT+KL), MEND (Mitchell et al., 2022a), ROME (Meng et al., 2022) and MEMIT (Meng et al., 2023), and (2) the addition-based methods: SERA (Mitchell et al., 2022b) and T-Patcher (Huang et al., 2023).

We use T-Patcher and ROME as the baseline editors to combine with our framework, denoting them as RASE-Patcher (R-Patcher) and RASE-ROME (R-ROME) respectively. We edit on Bert-base (Devlin et al., 2019), Bart-base (Lewis et al., 2020) provided by Meng et al. (2022), and GPT-2 XL[1] models. For R-Patcher, we insert five neurons to the last FFN layer on the FEVER and ZsRE datasets. We training the fact and sentence Encoder with a minibatch consists of 64 facts. For each fact, we select two consistent sentences as positive examples, other facts and sentences related to other facts

---

[1]https://huggingface.co/gpt2-xl

serve as negative examples. The threshold in Eq.4 is $t = 0.9$, $t_d = 0.15$ and $t_s = 0.05$. The Top-K used in Two-Way score is 5. Other parameters, like learning rate, will be set as those in the ROME and T-Patcher. All of our experiments were implemented on a single NVIDIA A100 GPU. See more details in Appendix E.

## 4.3 Experimental Results

### 4.3.1 Editing Small scale Models

Table 1 and Table 2 present the editing results on small scale models with R-Patcher. We first evaluate RASE on small number of edits.

Table 1: Results on small number of edits. R-Patcher is RASE-Patcher, +Pt is the results after we use the fact information, +Eq denotes that we add the equivalent inputs during editing based on R-Patcher+Pt .

| Editor | ZsRE (N=140) | | | | |
| --- | --- | --- | --- | --- | --- |
| | SR | ER | GR | TrainR | TestR |
| FT(last) | 1.000 | 0.300 | 0.580 | 0.914 | 0.924 |
| FT(last)+KL | 1.000 | 0.280 | 0.570 | 0.923 | 0.933 |
| MEND | 0.410 | 0.000 | 0.370 | 0.000 | 0.000 |
| SERA | 1.000 | 0.980 | 0.900 | 0.906 | 0.901 |
| T-Patcher | **1.000** | 0.990 | 0.820 | **0.997** | **0.996** |
| R-Patcher | **1.000** | 0.978 | 0.877 | 0.953 | 0.973 |
| +Pt | **1.000** | **1.000** | **0.976** | 0.976 | 0.964 |
| +Eq | **1.000** | **1.000** | **0.992** | 0.977 | 0.968 |

| Editor | FEVER (N=60) | | | | |
| --- | --- | --- | --- | --- | --- |
| | SR | ER | GR | TrainR | TestR |
| FT(last) | 1.000 | 0.590 | 0.610 | 0.893 | 0.946 |
| FT(last)+KL | 1.000 | 0.450 | 0.530 | 0.968 | 0.998 |
| MEND | 0.040 | 0.060 | 0.030 | 0.349 | 0.652 |
| SERA | **1.000** | **1.000** | 0.890 | 0.904 | 0.916 |
| T-Patcher | **1.000** | **1.000** | 0.820 | **0.999** | **1.000** |
| R-Patcher | **1.000** | **1.000** | 0.902 | 0.961 | 0.990 |
| +Pt | **1.000** | **1.000** | **1.000** | 0.961 | 0.986 |
| +Eq | **1.000** | **1.000** | **1.000** | 0.945 | 1.013 |

**RASE significantly improves editing generalization.** Table 1 illustrates the results for a small number of edits. Our method demonstrates competitive performance across two datasets and five metrics. It is noteworthy that the generalization (GR) is improved when guided by factual information (+Pt). We further enhance model generalization by incorporating consistently sampled data(+Eq) into the loss calculation during editing.

**RASE can maintain stable performance on more edits.** Table 2 illustrates the results for a large number of modifications. In comparison to Table 1, other methods experience a decline in performance as the number of edits increases. However, RASE

maintains consistent performance while demonstrating an advantage in generalization. We also test RASE on ZsRE with 4500 edit conditions, further validating its stability and sustainability. In contrast, T-Patcher's efficiency decreases as more modifications are made due to the continual addition of parameters to the language model. It becomes challenging for T-Patcher to perform large-scale consecutive edits within a certain time. For SERA, as the number of editing increases, its accuracy in discriminating modified data decreases, resulting in poor performance on TrainR and TestR. And MEND is a hyper-network editing method that predicts parameter changes for modifying current data by learning gradients of editing data. The predictions of MEND are highly dependent on the parameters of LMs, while continuous editing leads to constant parameter changes in LMs, rendering MEND ineffective for achieving sequential modifications. On the other hand, RASE maintains consistent efficiency even as the number of edit data increases.

Table 2: Results on large number of edits. N denotes the number of edits we edit, +ChatGPT means we use the ChatGPT to enhance our model (+Pt ), - means we use the $\mathbb{D}_{tr}$ as the edited dataset, so we did not calculate the retain score on $\mathbb{D}_{tr}$.

| Editor | ZsRE | | | | | |
| --- | --- | --- | --- | --- | --- | --- |
| | SR | ER | GR | TrainR | TestR | N |
| T-Patcher | 0.99 | **0.97** | 0.81 | 0.912 | 0.948 | 2766 |
| FT(all)+KL | 1.00 | 0.14 | 0.69 | 0.936 | 0.974 | 2766 |
| SERA | 1.00 | **0.97** | 0.90 | 0.728 | 0.694 | 2766 |
| R-Patcher | 1.00 | 0.95 | 0.863 | 0.973 | **0.975** | 2766 |
| +Pt | 1.00 | 0.95 | 0.93 | **0.976** | 0.964 | 2766 |
| +Eq | 1.00 | 0.96 | **0.97** | 0.976 | 0.965 | 2766 |
| +ChatGPT | 1.00 | 0.95 | 0.93 | **0.999** | **0.995** | 2766 |
| R-Patcher+Pt | 0.99 | 0.94 | 0.94 | - | 0.957 | 4502 |

| Editor | FEVER | | | | | |
| --- | --- | --- | --- | --- | --- | --- |
| | SR | ER | GR | TrainR | TestR | N |
| T-Patcher | 1.00 | 1.00 | 0.82 | **0.999** | 1.000 | 1231 |
| FT(all)+KL | 1.00 | 0.16 | 0.54 | 0.998 | **1.002** | 1231 |
| SERA | 1.00 | 1.00 | 0.89 | 0.717 | 0.709 | 1231 |
| R-Patcher | 1.00 | 1.00 | 0.92 | 0.854 | 0.972 | 1231 |
| +Pt | 1.00 | 1.00 | 0.93 | 0.867 | 0.973 | 1231 |
| +Eq | 1.00 | 1.00 | **0.97** | 0.867 | 0.974 | 1231 |
| +ChatGPT | **1.00** | **1.00** | **0.97** | 0.978 | 0.980 | 1231 |

**RASE can combine with large-scale language models such as ChatGPT to improve performance.** For the FEVER dataset, we achieve good results in modification performance (SR, GR, ER), but there is a decrease in TrainR. This is because numerous sentences are similar to the edits but dif-

fer by only a few words in FEVER. To address this, we employ ChatGPT to re-rank the Top-K results of retriever to evaluate if the input needs modification. We select the input that satisfies one of the three conditions in Eq.(4) as the hard data, construct the factual description of Top-K as a K-item decision task, and then let ChatGPT solve the problem. If there is no suitable answer, return *'None'*. Appendix D shows the usage of ChatGPT. The results with ChatGPT on both FEVER and ZsRE show that combining the results from the large model with our method can further enhancing the accuracy of fact retrieval and identify modified and unmodified data more accurately, thus maintaining the model's performance on the unedited data better.

### 4.3.2 Editing on large LMs

We use the GPT-2 XL (1.5B) to test the performance on a large-scale model.

Table 3: Editing Results on GPT-2 XL. FT-MEM means we include previously modified data and newly modified data as inputs for fine-tuning. R-ROME means we combine RASE with ROME. It is worth noting that in TrainR and TestR, some results exceeding 1 are due to the impact of modified data, which unintentionally corrects erroneous data that was not modified.

| Editor | ZsRE | | | | | |
|---|---|---|---|---|---|---|
| | SR | ER | GR | TrainR | TestR | N |
| FT | 0.515 | 0.093 | 0.070 | 0.261 | 0.247 | 1000 |
| FT-MEM | 0.454 | 0.489 | 0.175 | 0.356 | 0.341 | 1000 |
| MEND | 0.006 | 0.001 | 0.001 | 0.009 | 0.007 | 1000 |
| ROME | 0.997 | 0.520 | 0.378 | 0.620 | 0.572 | 1000 |
| MEMIT | 0.924 | 0.678 | 0.503 | **1.04** | 1.00 | 1000 |
| R-ROME | 0.997 | 0.966 | 0.657 | 1.002 | 1.004 | 1000 |
| R-ROME+pt | **0.997** | **0.972** | **0.754** | 1.036 | **1.007** | 1000 |
| FT | 0.447 | 0.095 | 0.073 | 0.461 | 0.421 | 5000 |
| ROME | 0.937 | 0.258 | 0.173 | 0.220 | 0.182 | 5000 |
| MEMIT | 0.433 | 0.001 | 0.001 | 0.001 | 0.001 | 5000 |
| R-ROME | 0.989 | 0.946 | 0.646 | 1.008 | 1.001 | 5000 |
| R-ROME+pt | **0.997** | **0.948** | **0.736** | **1.133** | **1.012** | 5000 |

**RASE can be flexibly combined with other editors and stably edit larger language models.** The results are shown in Table 3. When modifying 1000 data in ZsRE, ROME and MEMIT achieve impressive results, particularly in TrainR and TestR. However, FT and FT-MEM exhibit lower effectiveness, and MEND can not perform any modifications. Moreover, these methods display poor performance when evaluating the results of previous edits. In contrast, RASE preserves the modified results well without compromising the original model's performance. Even for TrainR, RASE is very close

to MEMIT. Although the overall performance in terms of generalization is relatively low, RASE demonstrates improvements compared to ROME. This further confirms that the fact-enhanced approach can enhance the generalization of different editors.

As the number of edits expanded to 5000, other methods exhibited a significant decline in effectiveness, particularly MEMIT. This is because MEMIT alters many parameters for each modification. As the edits accumulate, the original parameters of the model become disrupted, leading to the complete failure of the model. In contrast, RASE maintains favourable results and revalidating our approach's stability and sustainability.

### 4.4 Analysis

#### 4.4.1 Retrieval analysis

**Retrieval architecture analyse.** We compare different encoder architectures which are treated as retriever. The results are shown in Table 4. When directly using the Bart model as the encoder, the model struggles to identify which data needs modification. Izacard et al. (2022a) has achieved good results, but it has a lower HIT metric, indicating that it can successfully retrieve similar data but struggles to accurately identify some similar but different samples. After using RoBERTa with contrastive loss we proposed, there was a significant performance improvement. Furthermore, - w/o $L_{S2F}$ shows that the margin-based similarity can further enhance the HIT value and improving the overall recognition rate.

Table 4: Results on different encoder . **E&E** and **N&N** means the model successfully identifies the edits and unmodified data, respectively. **EBN** represents misidentifying edits as non-modified data and while **NBE** represents the opposite. **HIT**: represents the ratio of successful retrieval of the correct key value by the model.

| Model | E&E↑ | EBN↓ | NBE↓ | N&N↑ | **HIT**↑ |
|---|---|---|---|---|---|
| Bart-Base | 0.046 | 0.953 | **0.012** | **0.987** | 0.39 |
| Contriever | 0.914 | 0.085 | 0.071 | 0.928 | 0.936 |
| RoBERTa+$L_{cl}$ | **0.965** | **0.034** | 0.048 | 0.951 | **0.991** |
| - w/o $L_{S2S}$ | 0.955 | 0.044 | 0.037 | 0.962 | 0.989 |
| - w/o $L_{S2F}$ | 0.745 | 0.254 | 0.013 | 0.986 | 0.958 |
| - w/o $L_{S2S} + L_{S2F}$ | 0.615 | 0.384 | 0.079 | 0.920 | 0.971 |

**Relation for Retrieval augmented and Editing.** Knowledge editing (KE) is an arising and promising research area that aims to alter the parameters of some specific knowledge stored in pre-trained models so that the model can make new predictions

on those revised instances while keeping other irrelevant knowledge unchanged. Retrieval augmented models design retrieval strategies to incorporate world knowledge into a fixed LLM through prompting. Both of them are re-training free and have been shown to be effective for the issue of knowledge staleness.

However, Model editing methods prefer to edits LLM parameters directly, to update LM's knowledge, it is hard to maintain the other knowledge which are correct. Meanwhile, retrieval augmented models may not guarantee LMs can always update their predictions when given retrieved knowledge, due to the LLMs may prioritize their own parametric knowledge and ignores the retrieved non-parametric knowledge which be called knowledge conflict.

In this paper, we combine the advantage of retrieval augmented and editing methods. The retrieval module identifies the edits, and the editing module corrects the erroneous data. In our setting, the editing module processes only one data point at a time, so the retrieval module has a greater impact on ER, TrainR, and TestR, while the editing module has a greater influence on SR and GR.

### 4.4.2 Cost analysis

**Regarding editing efficiency**, we note that the efficiency of the existing continuous editing method, T-Patcher, gradually decreases as the number of edits increases. Our tests found that using T-Patcher to modify 1000 edits on an A100 GPU takes 2-3 days. On the other hand, RASE can perform edits at a stable speed. It takes approximately 50 seconds to modify a data. On GPT-2 XL, our efficiency remains consistent with ROME and MEMIT, but our method enables more sustained and reliable modifications. The time required to continuously modify 1000 edits is shown in Table 5.

Table 5: Time cost for different Editor on 1000 edits.

| Methods | Time |
|---------|--------|
| FT | 2 days |
| ROME | < 1 days |
| MEMIT | < 1 days |
| T-Patcher | 3 days |
| Ours | < 1 days |

**Regarding the Extra memory cost**, our retrieval augmented framework uses additional memory to store previous edits to achieve sequential

model editing. For R-Patcher, we save extra neurons for each edit, while for R-ROME, we save weight offsets for each edit. In T-Patcher, a portion of the training data is sampled to preserve locality to calculate memory loss. We estimate that storing 40,000 data would require approximately 120MB of space, and additional costs would be associated with maintaining the memory. Similarly, ROME also requires around 160MB of additional storage to maintain locality. In contrast, we use memory to save the previous edits. The cost for 1000 edits with R-ROME is 75MB, and 300MB for R-Patcher. Furthermore, the increase in memory usage is a fixed cost that does not increase with the number of edits.

**Regarding the Editing layer**, we follow the same settings as T-Patcher and ROME regarding the editing layer. For T-Patcher, we edit the FFN (Feed-Forward Network) in the last layer of the Transformer. For ROME, in GPT2-XL, we edit the FFN layer in the 17th layer. However, as Hase et al. (2023) suggests that *"Many layers could store a fact, and it happens that some do."* Therefore, there may be better choices for continuous editing than continuously modifying a specific layer. It would be beneficial to explore more flexible editing strategies by incorporating interpretability in the future.

### 4.4.3 Case study

Figure 4 gives a sense of how RASE would perform in practice, our approach can accurately modify X1-X3 without being affected by the content in memory K4, thus keeping X4 unchanged. On the hand, while T-Patcher can also correct X1-X3, it makes a mistake on X4, due to the influence of X3. This is why we adopt retrieval enhancement instead of directly modifying the model parameters.

## 5 Related Works

### 5.1 Knowledge Editing

Editing parametric knowledge is not as straight forward (Pan et al., 2023) as editing (Wang et al., 2009, 2010, 2014) knowledge graphs (Pan et al., 2017a,b). For editing parametric knowledge, a natural way is to use constrained fine-tuning to update parameters based on new examples (Zhu et al., 2020). However, in PLMs, a minor parameter change could change the model's predictions on many samples. Then, De Cao et al. (2021); Mitchell et al. (2022a); Han et al. (2023) trained a

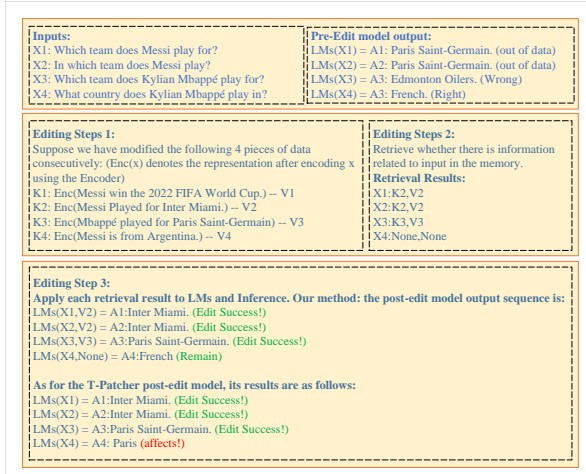

Figure 4: Example of Editing Steps.

Hyper-network to predict the parameter shift. Other methods like Meng et al. (2022, 2023) proposed a direct editing method and achieved great results on batch editing. More recently, some methods have developed external modules for edits and do not require access to base model parameters (Dong et al., 2022; Mitchell et al., 2022b). In order to apply the editing to real-world challenges better, Huang et al. (2023) proposed the Sequential Model Editing task and trained Transformer-Patcher, achieving edit the model up to a thousand times continuously.

## 5.2 Contrastive Learning

The key idea of contrastive learning (CL) is to pull semantically similar samples close and keep different samples apart (Hadsell et al., 2006; Chen et al., 2020). By employing contrastive learning objectives, Gao et al. (2021); Yan et al. (2021); Chuang et al. (2022); Zhou et al. (2022) fine-tuned the pretrained language models, resulting in significant advancements in learning unsupervised sentence embeddings. In order to alleviate the need for an annotated dataset, Gao et al. (2021); Liu et al. (2021) proposed a simple contrastive learning framework that used dropout noise within transformer layers to generate positive pairs. Nishikawa et al. (2022) proposed a contrastive learning method for learning sentence embeddings between sentences and their related entities sampled from Wikidata.

## 5.3 Retrieval-augmented language model

Retrieval augmentation can enhance language models' performance without significantly increasing the parameters and computation (Tirumala et al., 2022; Mialon et al., 2023). Khandelwal et al. (2020) increased PLMs memorization capacity by accessing memory units and an external look-up table. Borgeaud et al. (2022); Lazaridou et al. (2022); Izacard et al. (2022b) showed that retrieval improves performance across a variety of tasks such as question answering (Kwiatkowski et al., 2019), fact checking (Thorne et al., 2018b), dialogue (Dinan et al., 2019). Mitchell et al. (2022b) proposed a memory-based approach for knowledge editing. Inspired by retrieval, we view the editing task as a retrieval and augmentation process, construct a memory to store editing data, apply a certain modification through retrieval, and achieving stable and continuous editing. It is worth noting that, the editing is motivated by a list of bad cases of the form (question, wrong-answer, correct-answer), where the correct-answer is the knowledge that we mentioned above. Therefore, we only retrieve from all the facts related to the data that needs to be modified. And we assume that we have known all the erroneous and their corresponding correct answers.

## 6 CONCLUSION

This paper focuses on sequential model editing and proposes RASE, a retrieval augmented sequential model editing framework, to enhance the performance of existing editors in a plug-and-play manner, and achieve efficient, stable, and expansible Sequence Model Editing. We construct a fact-patch memory in a self-supervised way and utilize the memory to enhance the model's continuous editing capability. During editing, we use fact information related to the modified data as prompt to enhance the generalization of the editor. RASE has achieved favourable results under different scales of language models and varying numbers of edits. Additionally, it can be flexibly applied to different editors and integrating with large language models like ChatGPT can further enhance editing performance.

In the future, on the one hand, we plan to investigate knowledge editing for some complex tasks, such as reasoning. And explore how to integrate retrieval methods with model editing better. On the other hand, we might look into the connections between editing parametric knowledge and knowledge editing for uncertain knowledge graphs (Pan et al., 2005; Stoilos et al., 2006; Pan et al., 2007; Qi et al., 2007; Şensoy et al., 2013).

## Acknowledgements

This work has been supported by the Science and Technology Cooperation and Exchange Special Project of ShanXi Province (No.202204041101016), by the National Natural Science Foundation of China (No.62076155), by the Key Research and Development Program of Shanxi Province (No.202102020101008), and by the Chang Jiang Scholars Program (J2019032).

## Limitations

Our method focuses on editing factual knowledge, which is relatively easy to formalize and evaluate. Future work need to develop universal approaches that can edit all kinds of knowledge such as language and common sense knowledge in the same way. Additionally, the current metric for judging successful modifications is limited to whether the current input has been corrected. However, determining whether the model understands the underlying facts remains challenging and requires more rigorous evaluation metrics.

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

## A Contrastive Learning for Embedding

To learn sentence and fact representation, we training two Encoders $Enc_f(\cdot)$ and $Enc_s(\cdot)$ to get the fact embedding $E_f$ and sentence embedding $E_s$ respectively. We construct a minibatch containing $L$ facts, for each fact $f_i$, there are $S$ corresponding positive sentence $s_i{}^j$, $i \in [1, L]$, $j \in [1, S]$ [1], and any sentence unrelated to $f_i$ and any fact other than $f_i$ serve as negative examples for $f_i$. The input batch is denoted as $X = [f_1, s_1{}^1, s_1{}^2, ..., s_1{}^S, ..., f_L, s_L{}^1, s_L{}^2, ..., s_L{}^S]$. We train $Enc_f(\cdot)$ and $Enc_s(\cdot)$ using three contrastive losses and use RoBERTa (Liu et al., 2019) as the baseline PLMs. For convenience, we use $f$ and $s$ denote the Embedding $E_f = Enc_f(f)$ and $E_s = Enc_s(s)$ respectively:

**Facts Contrastive Learning (F2F)** aims to learn fact embedding in an unsupervised way. Given two embedding with different dropout masks $f_i$ and $f_i^+$, the training loss is:

$$\text{Loss}_{F2F} = -log \frac{e^{sim(f_i, f_i^+)/\tau}}{\sum_{j=1}^{L} e^{sim(f_i, f_j)/\tau}}, \quad (6)$$

where $\tau$ is a temperature hyper-parameter, and $sim(\cdot)$ is the cosine similarity.

**Sentences Contrastive Learning (S2S)** aims to learn sentence embedding in an unsupervised way. Since there are $S$ sentences in the same batch which are both related to the same $f_i$, we treat $S$ sentences as a group and learn sentence representations by the loss:

$$\text{Loss}_{S2S} = -log \frac{e^{sim(s_g, s_g^+)/\tau}}{\sum_{j=1}^{L} e^{sim(s_g, s_j)/\tau}}, \quad (7)$$

where $s_g = Pool(s_i^1, s_i^2, ..., s_i^S)$ is the mean representation of a group of inputs related to the same factual $f_i$, $Pool(\cdot)$ is the mean-pooling function, and $s_g^+$ is the dropout represents about $s_g$.

**Contrastive Learning between Sentence and Fact (S2F)** pulls the embedding of inputs and their related fact close while keeping unrelated facts apart. We use a margin-based similarity to increase the similarity between negative samples and weaken the similarity between positive samples, to ensuring that the similarity of positive samples is

higher than a certain threshold:

$$\text{Loss}_{S2F} = -log \frac{e^{\text{Ps}(s_i^*, f_i, m)}}{e^{\text{Ps}(s_i^*, f_i, m)} + \sum_{j \neq i}^{L} e^{\text{Ns}(s_i^*, f_j, n)}}, \quad (8)$$

where $* \in [1, S]$, $m, n$ is the margin value for positive and negative pairs. $\text{Ps}(\cdot)$ and $\text{Ns}(\cdot)$ are denoted as the positive and negative similarity scores respectively.

$$\text{Ps}(s_i^*, f_i, m) = (cos(s_i^*, f_i) - m)/\tau \quad (9)$$

For negative pairs, we will them a punishment when $h = cos(s_i^*, f_i) - cos(s_i^*, f_j)$ less than $m$:

$$\text{Ns}(s_i^*, f_j, n) = \begin{cases} (cos(s_i^*, f_j) + n)/\tau, & h < m, \\ cos(s_i^*, f_j)/\tau, & else. \end{cases} \quad (10)$$

In summary, our total loss for Contrastive Learning is:

$$\text{Loss}_{cl} = \lambda_1 \text{Loss}_{F2F} + \lambda_2 \text{Loss}_{S2S} + \lambda_3 \text{Loss}_{S2F}, \quad (11)$$

where $\lambda_i$ is hyper-parameters.

We train the encoder separately on ZsRE and FEVER datasets. It is worth noting that the dataset used for training the encoder and the dataset used for editing are sampled separately. And these encoders are pre-trained before the editing process. During editing, we utilize these encoders only. The $L$ is 64, and the maximum value of $S$ is 5.

## B Detail of the base editor

**Transformer-Patcher** In this paper, we follow (Huang et al., 2023), use Transformer-Patcher as a based editor because it only requires training a certain number of neurons for each edit and inserting them at a designated layer of the Transformer. In contrast to Huang et al. (2023), the patch used in our paper does not require additional memory to store training data to satisfy locality. Moreover, to enhance the generalization of the editor, we propose a fact-guided editor. We concatenate the factual information obtained through retrieval with the input data as a prompt to improve the generalization of the editing process.

For an edit $(x_e, y_{x_e})$ and the fact describe $F_{x_e}$, the input for the edit is $X_e = [x_e; F_{x_e}]$, and then we add few neurons to the FFN of special Transformer layers to alter the output of the edit. Formally, denote the hidden state of $X_e$ is $h$, for a standard

---

[1] As shown in Figure 3, the fact is "Iphone 14 is developed by Apple Inc.", and the postive sentence is "Iphone 14 was created by?", the negative sentence is "What club does Messi play for?".

FFN in Transformer blocks, the output of FFN($h$) is calculated by:

$$a = \text{Act}(h \cdot \mathbf{K} + b_k), \quad (12)$$

$$\text{FFN}(h) = a \cdot \mathbf{V} + b_v, \quad (13)$$

where $\text{Act}(\cdot)$ is a non-linear activation function such as Relu, $\mathbf{K} \in \mathbb{R}^{d_1 * d_2}$ and $\mathbf{V} \in \mathbb{R}^{d_2 * d_1}$ are the weight matrix of two linear in FFN respectively, $b_k$ and $b_v$ are two bias vectors. After add extra neurons ($\mathbf{K_n} \in \mathbb{R}^{d_1 * n}, \mathbf{v_n} \in \mathbb{R}^{n * d_2}, b_n$) in the FFN, the new output of $\text{FFN}_n(h)$ is calculated by:

$$[a \ a_n] = \text{Act}(h \cdot [\mathbf{K} \ \mathbf{k_n}] + [b_k \ b_k]), \quad (14)$$

$$\text{FFN}_n(h) = [a \ a_n] \cdot \begin{bmatrix} \mathbf{V} \\ \mathbf{v_n} \end{bmatrix} + b_v. \quad (15)$$

Substituting the Eq. (13) into Eq. (15) and calculating the following:

$$\text{FFN}_n(h) = \text{FFN}(h) + a_n \cdot v_n. \quad (16)$$

After training, we use target label $y_{x_e}$ and the edited output $u_{x_e}$ to calculate the loss $l_e$:

$$l_e = L(y_{x_e}, u_{x_e}), \quad (17)$$

where $L(\cdot)$ is a loss function (e.g. The Cross Entropy Loss).

**Rank-One Model Editing (ROME)** (Meng et al., 2022) applies a rank one edit to the down-projection matrix in a MLP layer in the model. It views the linear operate in Transformer with parameter $W$ as a key-value store for a set of vector keys $K = [k_1, k_2, ...k_n]$ and corresponding vector values $V = [v_1, v_2, ..., v_n]$, by solving $WK \approx V$. Each $k_i$ and $v_i$ denote a question and answer, respectively. Then for a new key-value pair $(k_*, v_*)$, we can insert this new data into the target layer by solving an optimization:

$$\text{minimize} ||\hat{W}K - V||,$$
$$such \ that \ \hat{W}k_* = v_*, \quad (18)$$
$$by \ setting \ \hat{W} = W + \Lambda(C^{-1}k_*)^T,$$

where $W$ is the weight matrix for the original linear, $C = KK^T$ is s a constant that we pre-cache by estimating the uncentered covariance of $k$ from a sample of Wikidata text, and $\Lambda = (v_* - Wk_*)/(C^{-1}k_*)^T k_*$ is a vector proportional to the residual error of the new key-value pair on the original memory matrix. $k_*$ is the average value over a small set of texts ending with the subject $s$, and $v_*$ is learning by optimization. In this paper, we use the ROME as one of the base editors because it does not need extra training and editing each data efficiently.

## C  Memory maintenance and usage

To further enhance the efficiency of memory retrieval, we employ the following methods to maintain the memory.

*1) Adding*: When a new edit has never appeared in the memory, we store the corresponding key-value pair. The key is the fact embedding about the edit and Encoder by the fact encoder $Enc_f(\cdot)$, and the value is the editing operation and the fact description for the edit.

*2) Updating*: When the key of a new edit is similar or identical to an existing key in the memory $M$, we merge the corresponding modification data with the existing one in the memory and train them together.

After the facts-patch memory is constructed, we will use the query module (section 3.2) to judge if each input needs to be edited, If not, the input will calculate by the original LMs. Once the input is the edit, we will apply the corresponding value to LMs temporarily. For T-Patch, we insert the trained extra neurons to the last layer's FFN of Transformer; for ROME, we use the $k_*$ and $v_*$ to calculate the parameter shift $\triangle W$ and replace the LMs parameters $W$ of layer $l$ by $\widehat{W} = W + \triangle W$. Once the current edit computation is completed, the model will be reset. Therefore, when we can successfully identify which data requires modification, our method ensures that the performance of the model on unmodified data remains unaffected.

## D  Constructing questions through ChatGPT

For an input $s$: "Brigitte Macron is married to someone who is President of the French Republic." we use our score function (cf.(3)) to select the Top-K facts, the result is:

$Facts$ = ["Brigitte Macron is the fiancee of Emmanuel Macron.", "Brigitte Macron is engaged.", "Brigitte Macron was born on April 23, 1953.", "Peggy Sue Got Married is a 1933 American film."] with the similar score $[0.939, 0.917, 0.731, 0.359, 0.354, 0.346]$.

However, the fact "Brigitte Macron is the fiancee of Emmanuel Macron." is not equivalent to "Brigitte Macron is married to someone who is President of the French Republic.", so our method should not edit this input. However, due to the similar high score, our query function chooses the first fact as the key and edits this data, influencing the result of the language model.

For this instance, we constructed a multiple-choice question as a prompt:

**Question**: Which of the following sentence expresses the same meaning as the sentence "Brigitte Macron is married to someone who is President of the French Republic.", If there is no answer, reply "None".

**Option**:

A): Brigitte Macron is the fiancee of Emmanuel Macron.

B): Brigitte Macron is engaged.

C): Brigitte Macron was born on April 23, 1953.

D): Peggy Sue Got Married is a 1933 American film.

Then, we utilize ChatGPT to answer the question and combine the answer as retrieval results with our framework, enabling a better assessment of whether the data needs modification.

## E   Details of Experiments setting

We evaluate our models on fact-checking dataset FEVER (Thorne et al., 2018a) and Zero-Shot Relation Extraction (ZsRE) dataset (Levy et al., 2017). We apply the BERT-base model Devlin et al. (2019) at the FEVER dataset. For ZsRE, we apply the BART-base model Lewis et al. (2020). We use the same data splits for both datasets as Huang et al. (2023). We use the original validate and employ it as $\mathbb{D}_{test}$ and the original $\mathbb{D}_{Train}$ is split into an edited set $\mathbb{D}_{edit}$, a new training set $\mathbb{D}'_{train}$ and a new validation $\mathbb{D}_{val}$ in a ratio (0.8:0.8:0.1 for FEVER and 0.9:0.075:0.025 for ZsRE).

We denote the initial model as $f_0$ which is trained on $\mathbb{D}'_{train}$ and validated on $\mathbb{D}_{val}$. Then the model $f_0$ is sequentially edited on $\mathbb{D}_{edit}$. For FEVER, the accuracy of the initial model is 87.6% on the edited dataset, 94.6% on the train dataset, and we get 10496 instances for the edited dataset; the mistake data is about 1300. For ZsRE, the accuracy of the initial model is 47.1% on the edited dataset and 56.9% on the training dataset; as a result, we get 5352 instances for the edited dataset, and the mistake data is about 2800. For both datasets, we randomly sampled a subset from $\mathbb{D}'_{train}$ with the size of 10,000 as $\mathbb{D}_{tr}$, and we training the fact Encoder and sentence Encoder on $\mathbb{D}'_{train}$ without $\mathbb{D}_{tr}$ and validate on $\mathbb{D}_{val}$.

And suppose there are $T$ mistakes in $\mathbb{D}_{edit}$, $I$ represents the indicator function, after editing the $t$-th edit example $(x_t, y_{x_t})$, we obtain a post-edit model $f_t$, and we use the following rate to evaluate our method.

**Success Rate** (SR): For each edit, we test if the post-edit model $f_t$ outputs the desired prediction:

$$SR = \frac{1}{T} \sum_{t=0}^{T} I(f_t(x_t) = y_{x_t}). \quad (19)$$

**Edit Retain Rate** (ER): After edited $T$ edits, we evaluate how many past edits are retained by the final model $f_T$:

$$ER = \frac{1}{T} \sum_{t=0}^{T} I(f_T(x_t) = y_{x_t}). \quad (20)$$

**Generalization Rate** (GR): After edited $T$ edits, we evaluate if $f_T$ is edited success on the equivalent dataset of the edit example in $\mathbb{D}_{edit}$:

$$GR = \frac{1}{TN_t} \sum_{t=0}^{T} \sum_{i=0}^{N_t} I(f_T(x_t^i) = y_{x_t^i}), \quad (21)$$

where $N_t$ is the number of the $t$-th edit equivalent input.

**Training Retain Rate** (TrainR): After edited $T$ edits, we compare the performance of the final model of $f_T$ and the initial model $f_0$ on subdataset $\mathbb{D}_{tr}$ which is randomly sampled from $\mathbb{D}'_{train}$.

$$TrainR = \frac{\sum_{(x,y)\in\mathbb{D}_{tr}} I(f_T(x) = y)}{\sum_{(x,y)\in\mathbb{D}_{tr}} I(f_0(x) = y)}. \quad (22)$$

**Test Retain Rate** (TestR): After edited $T$ edits, we compare the performance of the final model of $f_T$ and the initial model $f_0$ on original validate dataset $\mathbb{D}_{test}$.

$$TrainR = \frac{\sum_{(x,y)\in\mathbb{D}_{test}} I(f_T(x) = y)}{\sum_{(x,y)\in\mathbb{D}_{test}} I(f_0(x) = y)}. \quad (23)$$

Our baselines include:

- Fine-Tuning (FT) (Zhu et al., 2020): Directly fine-tunes the model on the edit example.

- FT with KL divergence (FT+KL) (Zhu et al., 2020): Fine-tunes the model on the edit example with an extra Kullback-Leibler (KL) constrained.

- MEND (Mitchell et al., 2022a): Using a hypernetwork to learn a parameter shift and then apply it to the model.

- SERA (Mitchell et al., 2022b): A variant of a memory-based model editor, which is provided by Huang et al. (2023).

- ROME (Meng et al., 2022): A Locate and Edit method for decoder-only models.

- MEMIT (Meng et al., 2023): A method extension of ROME that modifies the MLP weights of a range of critical layers.

- T-Patcher (Huang et al., 2023): A sequential editing method which adds and trains a few neurons in models.

```
input:          "What is the date of birth for Cari Lekebusch?"
▼ output:
    0:          "1972"
▼ rephrases:
    0:          "What's the date of birth of Cari Lekebusch?"
    1:          "What is the date of birth of Cari Lekebusch?"
    2:          "What is Cari Lekebusch's date of birth?"
    3:          "......"
  fact_rep:     "Cari Lekebusch, date_of_birth, 1972"
  subject:      "Cari Lekebusch"
▼ fact_rep_use: "Cari Lekebusch || When did Cari Lekebusch get born? || 1972"
```

Figure 5: Case in ZsRE.

## F  Dataset Samples

Figure 5 shows an example in ZsRE. For the triplet information involved in the question "<Cari Lekebusch, $date\_of\_birth$, 1972>", we convert it into a similar question format "Cari Lekebusch || When did Cari Lekebusch get born? || 1972" and use it as a factual description. When using facts to enhance generalization, we remove the labels: "Cari Lekebusch || When did Cari Lekebusch get born?".

Figure 6 shows an example in FEVER. The structure for FEVER is similar to ZsRE, but the difference is that we construct $fact\_rep\_use$ based on the statements and actual facts from FEVER. e.g. For the fact: <Amerigo Vespucci, $place\_of\_birth$, Italian>, we use the fact represent: "Amerigo Vespucci || Amerigo Vespucci was Spanish. || False", due to the claim is "Amerigo Vespucci was Spanish.".

```
id:             "40108"
input:          "Amerigo Vespucci was Spanish."
▼ output:
  ▼ 0:
      answer:   "REFUTES"
▼ rephrases:
    0:          "Amerigo Vespucci was in Spanish."
    1:          "Amerigo Vespucci was Hispanic."
    2:          "Amerigo Vespucci was a Spanish."
    3:          "......"
  fact_rep:     "Amerigo Vespucci , place_of_birth, Italian"
  subject:      "Amerigo Vespucci"
▼ fact_rep_use: "Amerigo Vespucci || Amerigo Vespucci was Spanish. || False"
```

Figure 6: Case in FEVER.