# OpenReview forum: "Improving Sequential Model Editing with Fact Retrieval"
_EMNLP/2023/Conference — EMNLP 2023 Findings_

### Official Review · Reviewer_Gyuf · 2023-07-29

**Soundness:** 4

**Excitement:**

3: Ambivalent: It has merits (e.g., it reports state-of-the-art results, the idea is nice), but there are key weaknesses (e.g., it describes incremental work), and it can significantly benefit from another round of revision. However, I won't object to accepting it if my co-reviewers champion it.

**Paper Topic And Main Contributions:**

This paper focuses on the task of sequential model editing (SME), which aims to continuously fix a series of mistakes in a pre-trained language model (PLM) as they appear, while maintaining performance on previous edits and unmodified data. The main contributions are:
- Proposes a retrieval augmented framework that stores edits and related factual information in a "fact-patch memory". A query module then retrieves from this memory to identify and apply edits.
- Introduces a fact-aware contrastive learning method to learn sentence and fact representations for retrieving related facts.
- Shows the framework can enhance existing model editors by providing relevant facts during editing.
- Demonstrates stable performance scaling to thousands of edits on FEVER and ZsRE datasets.

The problem of continuously updating PLMs without damaging overall performance is practically important as these models are deployed in products. The proposed retrieval approach is novel and the results demonstrate clear improvements in efficiency and scalability over prior SME methods.


**Questions For The Authors:**


- Could you provide any analysis on more real-world PLM mistakes to give a better sense of how the approach would perform in practice?
- Are there any further ablation studies you could do to better understand the source of improvements - is it the retrieval primarily or the fact representations?
- How does performance scale beyond the thousands of edits tested? Where do you see the limits?
- Is related factual knowledge always needed for good performance? How critical is access to this information?

**Reasons To Accept:**


- The paper addresses an important problem of efficiently editing factual mistakes in PLMs, which is very relevant as these models are deployed more widely.
- The proposed retrieval augmented framework is novel and intuitive. Storing prior edits for later retrieval is a sensible approach.
- The method clearly outperforms prior SME techniques in terms of efficiency, scalability, and edit retention on the FEVER and ZsRE datasets. The gains are substantial.
- The ablation studies provide useful insights into the impact of different components like the contrastive learning and integration with ChatGPT.
- The paper is clearly written and easy to follow overall. The background covers relevant prior work.


**Reasons To Reject:**


- The datasets used are somewhat artificial. More analysis on real-world mistakes could better highlight the benefits.
- There could be more ablation or analysis to understand exactly where the gains come from - is it mainly the retrieval or the fact representations?
- The efficiency gains flatten out after thousands of edits, suggesting there may be limits to scaling further. More discussion of limitations would be helpful.
- The approach relies on access to related factual knowledge, which may not always be available in practice. How critical is this knowledge?
- The gains over prior SME methods, while substantial, are a bit incremental. The novelty may not seem as high.


**Reproducibility:**

4: Could mostly reproduce the results, but there may be some variation because of sample variance or minor variations in their interpretation of the protocol or method.

**Reviewer Confidence:**

3: Pretty sure, but there's a chance I missed something. Although I have a good feel for this area in general, I did not carefully check the paper's details, e.g., the math, experimental design, or novelty.

---

> ### Author Rebuttal · Authors · 2023-08-28
>
> Thanks for your suggestions and comments!
>
> **Q1. The datasets used are somewhat artificial. More analysis on real-world mistakes could better highlight the benefits.**
>
> The datasets chosen (ZsRE and FEVER) are commonly used for editing tasks, with FEVER based on Wikipedia pages and reflects concepts in the real world. Maybe the example presented below in Q6 could help highlight some of the benefits.
>
> **Q2. There could be more ablation or analysis to understand exactly where the gains come from - is it mainly the retrieval or the fact representations?**
>
> Our editing model consists of two core modules: the retriever and the editor. They are responsible for different metrics. In general, the retriever has a greater impact on the metrics, as it determines whether the current input needs to be modified. If modification is necessary, the corresponding changes are applied to the model. Consequently, when data that doesn't require modification is mistakenly identified as needing modification, the TrainR and TestR are lowered. Similarly, when data that needs modification is wrongly classified as not needing modifications, the ER and GR are decreased. On the other hand, the SR metric indicates the success of the editor in modifying data that requires changes. Therefore, the SR is solely related to the performance of the editor.
>
> **Q3. The efficiency gains flatten out after thousands of edits, suggesting there may be limits to scaling further. More discussion of limitations would be helpful.**
>
> One of our contributions is to increase the upper limit of sequential editing while maintaining performance. Experimental results have also shown that our approach yields favorable results when modifying around 5,000 edits. In contrast, other methods exhibit poor performance, low efficiency, or entail significant editing costs.
> Regarding the "limits to scaling", we believe they still exist. The primary reason is that with the growing number of edits, instances with high similarity yet distinct differences could populate the memory. This situation impairs the performance of the retriever, consequently restricting editing upper. There can be different approaches to addressing this but they are clearly outside the scope of the paper. In the camera ready version, we will further clarify this issue in the Limitations section.
>
> **Q4. The approach relies on access to related factual knowledge, which may not always be available in practice. How critical is this knowledge?**
>
> This knowledge is critical as it is required as inputs according to the definition of  the Sequential Model Editing task (cf. Sec. 2.2). Intuitively, the editing is motivated by a list of bad cases of the form (question, correct-answer), where the corrct-answer is the knowledge that we mentioned above. We believe this task defiinition is rather close to real-world settings. If we do not have such knowledge, then it would be a different task.
>
> **Q5. The gains over prior SME methods, while substantial, are a bit incremental. The novelty may not seem as high.**
>
> In contrast to the previous SME approach, we are the first to adopt a retrieval-augmented strategy to achieve continuous model editing. Our method allows us to avoid modifying the original model parameters. As a result, while maintaining a certain level of performance, our method exhibits excellent reliability and stability. With the growing prevalence of large models, our method is more promising and better suited to these models.
>
> **Q6. Could you provide any analysis on more real-world PLM mistakes to give a better sense of how the approach would perform in practice?**
>
> Sure, in the following example, our approach can accurately modify X1-X3 without being affected by the content in memory K4, thus keeping X4 unchanged. On the hand, while T-Patcher can also correct X1-X3, it makes a mistake on X4, due to the influence of X3. This is why we adopt retrieval enhancement instead of directly modifying the model parameters.
>
> **Examples:**
> Assuming we have an input sequence:
>
> *X1: Which team does Messi play for?*
>
> *X2: In which team does Messi play?*
>
> *X3: Which team does Kylian Mbappé play for?*
>
> *X4: What country does Kylian Mbappé play in?*
>
> The original language model's output to X1, X2,X3 and X4 are as follows:
>
> *LMs(X1) = A1: Paris Saint-Germain. (out of data)*
>
> *LMs(X2) = A2: Paris Saint-Germain. (out of data)*
>
> *LMs(X3) = A3: Edmonton Oilers. (Wrong)*
>
> *LMs(X4) = A3: French. (Right)*
>
> Editing Steps:
> Step 1: Pre-Constructed memory containing 4 records (Key-Value Pairs), suppose we have modified the following 4 pieces of data consecutively (*Enc(x) denotes the representation after encoding x using the Encoder Enc.
> ):
>
> *K1: Enc(Messi win the 2022 FIFA World Cup.) -- V1*
>
> *K2: Enc(Messi Played for Inter Miami.) -- V2*
>
> *K3: Enc(Mbappé played for Paris Saint-Germain) -- V3*
>
> *K4: Enc(Messi is from Argentina.) -- V4*
>
> Step 2: Retrieve whether there is information related to input in the memory.
> Retrieval Results:
>
> *X1:K2,V2*
>
> *X2:K2,V2*
>
> *X3:K3,V3*
>
> *X4:None,None*
>
> Step 3: Apply each retrieval result to LMs and Inference.
> Our method: the post-edit model output sequence is:
>
> *LMs(X1,V2) = A1:Inter Miami. (Edit Success!)*
>
> *LMs(X2,V2) = A2:Inter Miami.  (Edit Success!)*
>
> *LMs(X3,V3) =A3:Paris Saint-Germain.  (Edit Success!)*
>
> *LMs(X4,None)=A4:French  (Remain)*
>
> As for the T-Patcher post-edit model, its results are as follows:
>
> *LMs(X1) = A1:Inter Miami.  (Edit Success!)*
>
> *LMs(X2) = A2:Inter Miami. (Edit Success!)*
>
> *LMs(X3) =A3:Paris Saint-Germain. (Edit Success!)*
>
> *LMs(X4)=A4: Paris (affects!)*
>
> **The answers to Q7, Q8, and Q9 have been provided in the response to Q2, Q3, and Q4.**

---

### Official Review · Reviewer_Eqru · 2023-08-03

**Soundness:** 3

**Excitement:**

3: Ambivalent: It has merits (e.g., it reports state-of-the-art results, the idea is nice), but there are key weaknesses (e.g., it describes incremental work), and it can significantly benefit from another round of revision. However, I won't object to accepting it if my co-reviewers champion it.

**Paper Topic And Main Contributions:**

This paper presents a retrieval-augmented editing framework to enhance the performance of existing editors in a plug-and-play manner. The authors first train a fact encoder and a sentence encoder in a self-supervised way. They construct a fact-patch memory and use a query module to classify the edits by retrieving related facts from the memory. The experiments validate the efficiency and generalization of this framework.

**Questions For The Authors:**

1. Why does the MEND have a poor performance in all experiments, especially on the ZsRE dataset?

2. Why don’t you use the ChatGPT to enhance your model on the ZsRE dataset?

**Reasons To Accept:**

1. This paper focuses on timely and sustainable correction and unknown to-be-modified data, which is different from previous work.

2. The proposed framework can be flexibly combined with other editors, showing its expandability.

**Reasons To Reject:**

1. The paper lacks some essential experiments. For example, more details of the editing speed and extra memory should be provided if the proposed method is different from previous methods on editing speed and extra memory.

2. Although the proposed method is interesting, some parts of this framework are not so novel, such as contrastive learning methods in the self-supervised encoder training module and memory construction.

3. The experiment results show that the proposed method has poor improvement. Actually, the proposed method gains a worse performance on some metrics.

4. The paper is hard to read, it could perhaps be alleviated by providing guided examples that help to understand.

**Reproducibility:**

4: Could mostly reproduce the results, but there may be some variation because of sample variance or minor variations in their interpretation of the protocol or method.

**Reviewer Confidence:**

4: Quite sure. I tried to check the important points carefully. It's unlikely, though conceivable, that I missed something that should affect my ratings.

---

> ### Author Rebuttal · Authors · 2023-08-28
>
> Thank you for your suggestions and comments!
>
> **Q1. The paper lacks some essential experiments. For example, more details of the editing speed and extra memory should be provided if the proposed method is different from previous methods on editing speed and extra memory.**
>
> While the main focus of our paper is to enhance the upper limit and stability of sequential model editing, we do provide a brief analysis in Section 4.3.1 paragraph 3 and Section 4.4 paragraph 2 on memory cost and editing efficiency. For your convenience, below are some further analysis, which we will supplement in the appendix of the camera ready version.
>
> **Regarding the Extra memory cost**, as discussed in section 4.4, to modify 3000 records, R-Patcher requires 300MB of storage space while T-Patcher needs 420MB, as T-Patcher needs 120MB for initialisation, which is not needed in R-Patcher. ROME needs only 160MB memory for editing, as it does not store the previous edits but rather update the parameters of the model directly.
>
> **Regarding editing efficiency**, in section 4.3.1, we note that the efficiency of the existing continuous editing method, T-Patcher, gradually decreases as the number of edits increases. Our tests found that using T-Patcher to modify 1000 edits on an A100 GPU takes 2-3 days. On the other hand, our method can perform edits at a stable speed. It takes approximately 50 seconds to modify a data. On GPT-2 XL, our efficiency remains consistent with ROME and MEMIT, but our method enables more sustained and reliable modifications. The time required to continuously modify 1000 edits is:
>
> | Methods | Time |
> | --- | --- |
> | T-Patcher | 3 days |
> | FT | 2 days |
> | ROME | < 1 days |
> | MEMIT | < 1 days |
> | Ours | < 1 days |
>
> **Q2. Although the proposed method is interesting, some parts of this framework are not so novel, such as contrastive learning methods in the self-supervised encoder training module and memory construction.**
>
> In contrast to the previous SME approach, we are the first to adopt a retrieval-augmented strategy to achieve continuous model editing. Through retrieval augmentation, we can:
>
> - Efficiently make sequential modifications to the model's knowledge while quickly adapting to models of various scales;
> - Avoid compromising the model's original performance and ensure controlled editing or modifications, which is crucial in applications.
>
> Additionally, the techniques we employ have been enhanced based on the editing tasks:
>
> - The retrieval under editing tasks requires a high demand for Top1(R@1) accuracy and the ability to identify semantically consistent data. Therefore, we incorporate margin-based similarity and group loss into the contrastive learning loss separately to enhance the performance of the retriever.
> - Existing memory-based methods give less consideration to the cost of memory maintenance. At the same time, we take the representation of modified data as the key and the change of modified parameters as a value. In this way, our method has the following advantages:
>     - Achieving editing in a plug-and-play manner.
>     - For frequently changing facts, we can still quickly locate and update this information, thus achieving cost-effective maintenance.
>
> **Q3. The experiment results show that the proposed method has poor improvement. Actually, the proposed method gains a worse performance on some metrics.**
>
> Our work primarily addresses the issue of the sustainability of Sequential model editing. As shown in Sec.4 Table 1 and Table 2, as the number of edits increases, our method continues to maintain stable performance and outperforms existing methods. Furthermore, our approach achieves breakthroughs in sustained editing numbers, which is one of the main objectives of this paper.
>
> The slightly “worse performance on some metrics” is mainly concentrated in a small number of edits, while our advantage lies in conditions involving substantial edits, as shown in Sec.4 Tables 2 and Table 3. As the amount of data being modified increases, the superiority of our method becomes increasingly apparent, and our overall metrics outperform those of existing methods.
>
> Furthermore, existing editing methods alter the model parameters and the inherent lack of interpretability of the model, the effects of even minor parameter perturbations are difficult to estimate. As a result, both TrainR and TestR are unreliable. In contrast, we achieve edits while keeping the model parameters unchanged, resulting in more reliable TrainR and TestR values.
>
> **Q4. The paper is hard to read, it could perhaps be alleviated by providing guided examples that help to understand.**
>
> Sure, in the following example, our approach can accurately modify X1-X3 without being affected by the content in memory K4, thus keeping X4 unchanged. On the hand, while T-Patcher can also correct X1-X3, it makes a mistake on X4, due to the influence of X3. This is why we adopt retrieval enhancement instead of directly modifying the model parameters.
>
> **Examples:**
> Assuming we have an input sequence:
>
> *X1: Which team does Messi play for?*
>
> *X2: In which team does Messi play?*
>
> *X3: Which team does Kylian Mbappé play for?*
>
> *X4: What country does Kylian Mbappé play in?*
>
> The original language model's output to X1, X2,X3 and X4 are as follows:
>
> *LMs(X1) = A1: Paris Saint-Germain. (out of data)*
>
> *LMs(X2) = A2: Paris Saint-Germain. (out of data)*
>
> *LMs(X3) = A3: Edmonton Oilers. (Wrong)*
>
> *LMs(X4) = A3: French. (Right)*
>
> Editing Steps:
> Step 1: Pre-Constructed memory containing 4 records (Key-Value Pairs),suppose we have modified the following 4 pieces of data consecutively (*Enc(x) denotes the representation after encoding x using the Encoder Enc.):
>
> *K1: Enc(Messi win the 2022 FIFA World Cup.) -- V1*
>
> *K2: Enc(Messi Played for Inter Miami.) -- V2*
>
> *K3: Enc(Mbappé played for Paris Saint-Germain) -- V3*
>
> *K4: Enc(Messi is from Argentina.) -- V4*
>
> Step 2: Retrieve whether there is information related to input in the memory.
> Retrieval Results:
>
> *X1:K2,V2*
>
> *X2:K2,V2*
>
> *X3:K3,V3*
>
> *X4:None,None*
>
> Step 3: Apply each retrieval result to LMs and Inference.
> Our method: the post-edit model output sequence is:
>
> *LMs(X1,V2) = A1:Inter Miami. (Edit Success!)*
>
> *LMs(X2,V2) = A2:Inter Miami.  (Edit Success!)*
>
> *LMs(X3,V3) =A3:Paris Saint-Germain.  (Edit Success!)*
>
> *LMs(X4,None)=A4:French  (Remain)*
>
> But the T-Patcher post-edit model results are as follows:
>
> *LMs(X1) = A1:Inter Miami.  (Edit Success!)*
>
> *LMs(X2) = A2:Inter Miami. (Edit Success!)*
>
> *LMs(X3) =A3:Paris Saint-Germain. (Edit Success!)*
>
> *LMs(X4)=A4: Paris (affects!)*
>
> **Q5. Why does the MEND have a poor performance in all experiments, especially on the ZsRE dataset?**
>
> MEND is a hyper-network editing method that predicts parameter changes for modifying current data by learning gradients of editing data. The predictions of MEND are highly dependent on the parameters of LMs, while continuous editing leads to constant parameter changes in LMs, rendering MEND ineffective for achieving sequential modifications.
>
> **Q6. Why don’t you use the ChatGPT to enhance your model on the ZsRE dataset?**
>
> We are using ChatGPT to leverage its semantic computing capability to assist in the retrieval process, thereby enhancing the quality of search results. We didn't report the use of ChatGPT to enhance the model on the ZsRE dataset because our method already large outperforms T-Patcher on the ZsRE dataset. To address your concerns, we report the results of ChatGPT on ZsRE as follow and will include them in the camera ready version.
>
> | N=2766 | SR | ER | GR | TrainR | TestR |
> | --- | --- | --- | --- | --- | --- |
> | T-Patcher | 0.99 | **0.97** | 0.81 | 0.912 | 0.948 |
> | R-Pathcer | 1.00 | 0.95 | 0.863 | 0.973 | 0.975 |
> | +Pt | 1.00 | 0.95 | 0.93 | 0.976 | 0.964 |
> | +ChatGPT | **1.00** | 0.95 | **0.93** | **0.999** | **0.995** |

---

### Official Review · Reviewer_z2e8 · 2023-08-05

**Soundness:** 4

**Excitement:**

5: Transformative: This paper is likely to change its subfield or computational linguistics broadly. It should be considered for a best paper award. This paper changes the current understanding of some phenomenon, shows a widely held practice to be erroneous in someway, enables a promising direction of research for a (broad or narrow) topic, or creates an exciting new technique.

**Paper Topic And Main Contributions:**

The paper is trying to improve the sequential model editing by retrieval based framework.
It defines a framework leveraging the self-supervised trained fact and sentence encoder to maximize the similarity among sentences and relevant fact description. Then, the Fact is used as key while the parameter shift is regarded as value. Given one input, it identifies a key and applies the parameter shift.
Experiments are conducted on fact checking dataset and relation extraction dataset. Experiment results can clearly demonstrate the strength of the approach.

**Reasons To Accept:**

The paper proposes a general framework/solution for the large sequential model editing.
Experiments have shown the method can be easily applied to other existing approaches.
In addition, the experiment results show the proposed model is so effective that the improvements are quite significant.
The paper is very well written with clear conclusion and methodology.


**Reasons To Reject:**

It would be better to consider various factual information like the information from knowledge graph. This can be a future work.


**Reproducibility:**

5: Could easily reproduce the results.

**Reviewer Confidence:**

3: Pretty sure, but there's a chance I missed something. Although I have a good feel for this area in general, I did not carefully check the paper's details, e.g., the math, experimental design, or novelty.

---

> ### Author Rebuttal · Authors · 2023-08-28
>
> Thank you very much for your suggestions, and we also appreciate your approval of our work!
>
> Integrating the knowledge graph with model editing is interesting. In the future, we hope to utilize the knowledge graph to achieve propagative edits and to extend model editing to richer types of knowledge.

---

### Meta-Review · Area_Chair_pJhP · 2023-09-18

**Recommendation:** 4

**Metareview:**

This paper studies the task of sequential Model editing (Huang+, 2023) -- fixing erroneous knowledge in LLMs continuously. The speed of existing editors requires continuous modification of LM params and has an increasing cost during sequential editing.
This work presents a retrieval-augmented editing framework. it borrows the memory-based editing approach (Mitchell+ 2022) and applies it to the continuous setting. It
1.  trains a fact encoder and a sentence encoder with self-supervised contrastive losses.
2.  construct a fact-patch memory
3. uses a query module then retrieves from this memory to identify and apply edits.

Experiments on FEVER and ZsRE datasets shows the framework can enhance existing model editors by providing relevant facts during editing. The approach scales to thousands of edits.

Strength:
1. The paper addresses an important problem of sequential model editing (although the paper has given little context of why this task is important in production settings).
2. The proposed retrieval approach is novel and the results demonstrate clear improvements in efficiency and scalability over prior SME methods
3. The proposed framework can be flexibly combined with other editors.

Weakness:
1. More details of the editing speed and extra memory should be prominently presented in the paper.
2. I think it would be helpful to discuss the relationship (and pros/cons) comparing SME to retrieval augmented models (e.g. REALM).
3. more ablation study is needed to explain where the gains come from (e.g., the retrieval vs the fact representations).

---

### Decision · Program_Chairs · 2023-10-07

**Decision:**

Accept-Findings

**Comment:**

This paper studies the task of sequential Model editing (Huang+, 2023) -- fixing erroneous knowledge in LLMs continuously. The speed of existing editors requires continuous modification of LM params and has an increasing cost during sequential editing.
This work presents a retrieval-augmented editing framework. it borrows the memory-based editing approach (Mitchell+ 2022) and applies it to the continuous setting. It
1.  trains a fact encoder and a sentence encoder with self-supervised contrastive losses.
2.  construct a fact-patch memory
3. uses a query module then retrieves from this memory to identify and apply edits.

Experiments on FEVER and ZsRE datasets shows the framework can enhance existing model editors by providing relevant facts during editing. The approach scales to thousands of edits.

Strength:
1. The paper addresses an important problem of sequential model editing (although the paper has given little context of why this task is important in production settings).
2. The proposed retrieval approach is novel and the results demonstrate clear improvements in efficiency and scalability over prior SME methods
3. The proposed framework can be flexibly combined with other editors.

Weakness:
1. More details of the editing speed and extra memory should be prominently presented in the paper.
2. I think it would be helpful to discuss the relationship (and pros/cons) comparing SME to retrieval augmented models (e.g. REALM).
3. more ablation study is needed to explain where the gains come from (e.g., the retrieval vs the fact representations).